# An α-Gal antigenic surrogate as a biomarker of treatment evaluation in *Trypanosoma cruzi*-infected children. A retrospective cohort study

**Manuel Abal[1,2], Virginia Balouz[1,2], Rosana Lopez[3], M. Eugenia Giorgi[3], Carla Marino[3], Cintia V. Cruz[4,5], Jaime Altcheh[4,6]\*, Carlos A. Buscaglia[1,2]\***

1 Instituto de Investigaciones Biotecnológicas (IIBio), Universidad Nacional de San Martín (UNSAM), and Consejo Nacional de Investigaciones Científicas y Técnicas (CONICET), Buenos Aires, Argentina, 2 Escuela de Bio y Nanotecnologías (EByN), UNSAM, Buenos Aires, Argentina, 3 Centro de Investigaciones en Hidratos de Carbono (CIHIDECAR), Departamento de Química Orgánica, Facultad de Ciencias Exactas y Naturales, Universidad de Buenos Aires (UBA), and CONICET, Buenos Aires, Argentina, 4 Servicio de Parasitología-Chagas, Hospital de Niños 'Dr Ricardo Gutierrez', and Instituto Multidisciplinario en Investigaciones Pediátricas (IMIPP) CONICET-GCBA, Buenos Aires, Argentina, 5 Mahidol Oxford Research Unit (MORU), Centre for Tropical Medicine and Global Health, Nuffield Department of Medicine, University of Oxford, United Kingdom, 6 Fundación para el estudio de las infecciones parasitarias y enfermedad de Chagas (FIPEC foundation), Buenos Aires, Argentina

\* jaltcheh@gmail.com (JA); cbuscaglia89@gmail.com (CAB)

**Data Availability Statement:** All relevant data are within the manuscript and its Supporting Information files.

## Abstract

### Background

Proper evaluation of therapeutic responses in Chagas disease is hampered by the prolonged persistence of antibodies to *Trypanosoma cruzi* measured by conventional serological tests and by the lack of sensitivity of parasitological tests. Previous studies indicated that tGPI-mucins, an α-Gal (α-D-Gal*p*(1→3)-β-D-Gal*p*(1→4)-D-GlcNAc)-rich fraction obtained from *T. cruzi* trypomastigotes surface coat, elicit a strong and protective antibody response in infected individuals, which disappears soon after successful treatment. The cost and technical difficulties associated with tGPI-mucins preparation, however, preclude its routine implementation in clinical settings.

### Methods/principle findings

We herein developed a neoglycoprotein consisting of a BSA scaffold decorated with several units of a synthetic α-Gal antigenic surrogate (α-D-Gal*p*(1→3)-β-D-Gal*p*(1→4)-β-D-Glc*p*). Serological responses to this reagent, termed NGP-Tri, were monitored by means of an in-house enzyme-linked immunosorbent assay (α-Gal-ELISA) in a cohort of 82 *T. cruzi*-infected and Benznidazole- or Nifurtimox-treated children (3 days to 16 years-old). This cohort was split into 3 groups based on the age of patients at the time of treatment initiation: Group 1 comprised 24 babies (3 days to 5 months-old; median = 26 days-old), Group 2 comprised 31 children (7 months to 3 years-old; median = 1.0-year-old) and Group 3 comprised 26 patients (3 to 16 years-old; median = 8.4 years-old). A second, control cohort (Group 4)

**Funding:** Research reported in this publication was supported by the Wellcome Trust (United Kingdom) under award number 222754/Z/21/Z (to JA) and by Agencia Nacional de Promoción de la Investigación, el Desarrollo Tecnológico y la Innovación (Agencia I+D+i, Argentina) under award numbers PICT-2017-I-A-3908 and PICT-2021-I-A-00284 (to CAB), PICT-2020-I-INVI-02396 (to VB) and PICT-2021-I-A-00441 (to CM). The funders had no role in study design, data collection and analysis, decision to publish, or preparation of the manuscript.

**Competing interests:** RL holds a fellowship from the Argentinean Research Council (CONICET) and VB, MEG, CM, JA and CAB are career investigators from the same institution. In addition, JA is a Clinical Investigator from the Hospital de Niños 'Dr Ricardo Gutiérrez' (Ministerio de Salud, Gobierno de la Ciudad de Buenos Aires).

included 39 non-infected infants (3 days to 5 months-old; median = 31 days-old) born to *T. cruzi*-infected mothers. Despite its suboptimal seroprevalence (58.4%), α-Gal-ELISA yielded shorter median time values of negativization (23 months [IC 95% 7 to 36 months] vs 60 months [IC 95% 15 to 83 months]; $p$ = 0.0016) and higher rate of patient negative sero-conversion (89.2% vs 43.2%, $p$ < 0.005) as compared to conventional serological methods. The same effect was verified for every Group, when analyzed separately. Most remarkably, 14 out of 24 (58.3%) patients from Group 3 achieved negative seroconversion for α-Gal-ELISA while none of them were able to negativize for conventional serology. Detailed analysis of patients showing unconventional serological responses suggested that, in addition to providing a novel tool to shorten follow-up periods after chemotherapy, the α-Gal-ELISA may assist in other diagnostic needs in pediatric Chagas disease.

## Conclusions/significance

The tools evaluated here provide the cornerstone for the development of an efficacious, reliable, and straightforward post-therapeutic marker for pediatric Chagas disease.

### Author summary

The limits of the current criterion for cure, i.e., negative seroconversion determined by conventional serology, and the lack of validated and sensitive markers for early assessment of response to trypanocidal drugs in Chagas disease stress the necessity of novel therapeutic response markers. Towards this goal, we herein developed by synthetic chemistry a neoglycoprotein bearing an α-Gal antigenic surrogate, termed NGP-Tri, and evaluated its performance in a large cohort of *T. cruzi*-infected and treated children. An α-Gal-ELISA built upon NGP-Tri showed a very good performance on the assessment of therapy efficacy, as it displayed significantly shorter median time value of negativization and higher rate of patient negative seroconversion as compared to conventional serology methods. These findings indicate that the tools developed here, as well as optimized versions derived from them, should have a positive impact on the clinical management of Chagas disease and the identification and/or clinical validation of novel drug candidates for the treatment of *T. cruzi* infections.

## Introduction

Chagas disease, caused by the protozoan parasite *Trypanosoma cruzi*, is a neglected, life-long disease for which no vaccines are yet available. With ∼6.5 million people already infected and up to 120 million individuals at risk of infection, it constitutes an important parasitic disease in Latin America; and also an emerging threat to global public health [1,2]. In endemic areas, *T. cruzi* transmission primarily occurs by exposure to the contaminated feces of blood-sucking tria-tomine vectors. However, humans can also become infected through the ingestion of tainted food/fluids, *via* blood transfusion, organ transplantation or transplacentally. According to epidemiological data, the latter mode of transmission occurs in ∼5% of babies born to *T. cruzi*-infected mothers, and accounts for up to 22% of new infections in Latin American countries [3].

Only two drugs, benznidazole (BZ) and nifurtimox (NF) are currently available for Chagas disease chemotherapy [4]. While these compounds are highly effective if administered in acute

disease, their efficacy in chronically infected individuals is lower and variable [4]. These drugs display adverse effects (i.e., weight loss, allergic dermatitis, etc.), which leads to up to $\sim$20–30% of patients permanently interrupting the treatment [5]. Moreover, BZ and NF cannot be used to treat pregnant women due to teratogenic risks [4]. In this framework, development of novel trypanocidal chemotherapies emerges as a main research priority [6–11].

A major limitation for drug discovery in Chagas disease is imposed by the lack of appropriate response biomarkers [12]. So far, the only accepted criterion of cure is negative conversion by conventional serological tests [13]. In the last decade, a consistent negative result in parasitological tests has also been used, mostly in clinical trials, as an efficacy biomarker. However, a significant proportion of patients are negative for parasitological techniques prior to treatment, thus making a subsequent negative result uninformative. Nevertheless, PCR-based methods are useful in certain clinical situations associated with patent blood parasitemia such as congenital infections, disease reactivation or therapeutic failure, but they cannot be used as a criterion of parasite elimination [12]. Conventional serological methods, i.e., enzyme-linked immunosorbent assay (ELISA) or indirect hemagglutination assay (IHA), are also hampered by intrinsic limitations. Due to the complexity of the 'antigen' used by these techniques (entire *T. cruzi* parasites or crude homogenates derived thereof), it may take years for patients to achieve negative seroconversion [14]. In addition, conventional serological techniques display low predictive value for diagnosis and/or follow-up of congenital infections due to the passive transfer of maternal antibodies [3].

Aiming at developing reliable post-therapeutic biomarkers, different strategies have been explored. These included host-derived biochemical and/or immunological signatures such as cytokine patterns, specific cellular responses and, mostly, antibodies to defined *T. cruzi* antigens or antigenic fractions [15–18]. Among the latter, the best results were obtained with the 'F2/3' or tGPI-mucins fraction, which is obtained by sequential solvent partitions from purified bloodstream trypomastigote forms, and which basically consists of highly *O*-glycosylated mucins [19,20]. tGPI-mucins reactivity is driven by α-Gal (α-D-Gal*p*(1→3)-β-D-Gal*p*(1→4)-α-D-GlcNAc) glycotopes that represent $\sim$10% of their complete glycan content [19]. In contrast to most mammals, humans do not express α-Gal due to the inactivation (i.e., pseudogenization by two single base deletions causing premature STOP codons) of the α1,3-galactosyltransferase gene [21]. As a result, human infections with *T. cruzi* or other pathogens bearing surface α-Gal glycotopes were shown to elicit strong and protective humoral responses against these structures [22–26]. It should be noted, however, that α-Gal antibodies may also be elicited in response to cross-reactive α-galactosyl-containing glycans displayed by commensal enterobacteria[27].

The tGPI-mucins demonstrated excellent sensitivity, specificity, and accuracy as a Chagas disease diagnostic biomarker[19,28]. In addition, antibodies to this fraction were shown to disappear from patients' circulation concurrently or soon after parasite elimination, thereby affording an appropriate marker of cure[29–33]. However, methodological drawbacks, i.e., need for culture of infective forms of the parasite, costly and difficult purification procedures, batch-to-batch inconsistencies, etc., preclude its routine implementation in clinical settings. As an alternative approach, the use of neoglycoproteins (NGPs) containing tGPI-mucins oligosaccharides has been proposed [6,34–38]. We have recently developed one NGP, henceforth NGP-Tri, consisting of a carrier protein (BSA) decorated with several units of the synthetic trisaccharide α-D-Gal*p*(1→3)-β-D-Gal*p*(1→4)-β-D-Glc*p* [39]. Serological characterizations showed that this trisaccharide is an α-Gal antigenic surrogate, as it is recognized by α-Gal antibodies from *T. cruzi*-infected individuals [39]. In this work, we evaluated NGP-Tri as a treatment efficacy biomarker in children with Chagas disease.

## Methods

### Ethics statement

The study protocol was approved by the research and teaching committee, and the bioethics committee from the Hospital de Niños 'Dr Ricardo Gutierrez'. Written informed consent was required from each patient's legal representatives as well as assent from the patient, if applicable. All samples were decoded and de-identified before they were provided for research purposes.

### Study population and screening for *T. cruzi* infection

A retrospective cohort of 82 children (3 days to 16 years-old at the time of treatment initiation) with diagnosis of *T. cruzi* infection at Servicio de Parasitología-Chagas, Hospital de Niños 'Dr Ricardo Gutierrez', Buenos Aires, Argentina, were recruited for this study. Eighty-one of them were treated and followed up following current normatives. *T. cruzi* infection in children under 8 months of age was assessed by a direct parasitological microhematocrit method [31]. In case of negative results, they were serologically tested at >8 months of age, upon clearance of passively transferred maternal antibodies [3]. *T. cruzi* infection in children over 8 months of age was diagnosed using two conventional serological tests: an ELISA that uses crude parasite homogenates (Chagatest-ELISA or tELISA) and an IHA (Chagatest-HAI; both from Wiener Laboratory, Rosario, Argentina). The control group consisted of 39 non-infected infants born to *T. cruzi*-infected mothers. All of them yielded microhematocrit negative results early after birth and consistent negative serological results at >8 months of age.

### Treatment

*T. cruzi*-infected children were treated with BZ 5–8 mg/kd/day or NF 10–15 mg/kg/day for 60 days [31]. Medication was provided in monthly batches, and compliance was assessed by tablet counting at each visit. Caregivers were also provided with a treatment diary to record doses administered, times of doses, symptoms, and problems associated with the treatment. Serum samples were taken at the time of diagnosis (pre-treatment), during treatment and after treatment. A detailed clinical history, physical examination, and laboratory routine tests were conducted during treatment, and conventional serology was carried out in every medical visit along the follow-up. In some cases, real-time PCR (qPCR) tests targeting satellite sequences in the *T. cruzi* genomic DNA were also carried out [12].

### Synthesis of Neoglycoproteins

Development of NGPs consisting of BSA covalently coupled to α-D-Gal$p$(1→3)-β-D-Gal$p$ (1→4)-β-D-Glc$p$ (NGP-Tri) or to D-Gal$p$ (NGP-Mono) has been described [39]. Briefly, protected *S*-tolyl glycosides of synthetic α-D-Gal$p$(1→3)-β-D-Gal$p$(1→4)-β-D-Glc$p$ or D-Gal$p$ were glycosidated with 6-benzyloxycarbonylamino-1-hexanol, using NIS/TfOH as promoter. After total deprotection, each glycoside was reacted with dimethyl squarate at pH 7 and then with BSA at pH 9 to afford NGP-Tri and NGP-Mono [40]. Characterization and quality assessment of NPGs was performed using SDS-PAGE and MALDI-TOF-MS. Both NPGs were homogeneous species bearing ∼29 units of α-D-Gal$p$(1→3)-β-D-Gal$p$(1→4)-β-D-Glc$p$ (NGP-Tri) or ∼35 units of D-Gal$p$ (NGP-Mono) per molecule of BSA [39].

## Enzyme-linked immunosorbent assay for detection of α-Gal antibodies (α-Gal-ELISA)

Final conditions of the α-Gal-ELISA test (amount of coating antigen per well, serum sample dilution and secondary antibody concentration) were established following checkerboard titration analysis, using NGP-Mono as negative control. Briefly, NGP-Tri was dissolved in carbonate buffer (pH 9.6) at 10 μg/mL, dispensed on flat-bottomed 384-well plates (Nunc, Roskilde, Denmark) at 30 μL/well and incubated ON at 4˚C. Plates were washed 3 times with PBS containing 0.05% Tween 20 (PBS/T) and blocked for 1 h at 37˚C in PBS/T supplemented with 4% skim milk. Serum samples were diluted 1:250 in a blocking buffer before being added to the plate. Following incubation for 1 h at 37˚C and extensive washings with PBS/T, peroxidase-conjugated goat IgG to human IgG/A/M immunoglobulins (Sigma) diluted 1:2,500 in blocking buffer was added to the plates and incubated at 37˚ C for 1 h. Plates were washed with PBS/T and incubated with 30 μL of citrate-phosphate buffer (pH 5) containing 0.2% hydrogen peroxide and 0.3 μL of 3,3',5,5'-tetramethylbenzidine (Sigma) prepared at 10 mg/mL in dimethylsulfoxide. The reaction was stopped with 5 μL of 2 M sulfuric acid, and the absorbance at 450 nm was read. Each sample was assayed in duplicate. Due to the variable content of α-Gal antibodies in human populations, different cutoff values were used. These cutoffs were established upon serum samples obtained from healthy children (under 1 year-old, from 1 to 10 years-old, and over 10 years-old) that rendered negative results for the aforementioned *T. cruzi* serological assays. When assaying serum samples from *T. cruzi*-infected mothers, cutoff values were established using samples from non-infected adults. A sample showing mean reactivity greater than the corresponding cutoff value (mean + 3 SD of the age-matched negative population), was considered positive. For comparison purposes, cutoff and sample values were normalized to a positive control (a serum from a chronic Chagas disease patient yielding 0.8 to 1.4 absorbance units) included in each assay. Selected samples were also evaluated for antibodies to TSSA (*T. cruzi* trypomastigote small surface antigen [41]). To that end, a glutathione *S*-transferase fusion protein bearing residues 24 to 62 of TSSAII was expressed in *Escherichia coli*, purified by glutathione affinity chromatography [42], and used as antigen in a previously described TSSA-ELISA [43].

## Statistical analysis

For serological regression analyses, reactivity values for tELISA and α-Gal-ELISA were expressed as a percentage of the corresponding pre-treatment sample. Reactivity of negative samples (determined as stated above) was expressed as zero. A linear regression model was used to examine the course of anti-α-Gal or anti-*T. cruzi* antibody levels over time. For each group of patients and each method, a slope parameter (with 95% confidence interval, CI) based upon time-point data for which at least one patient per group was positive, was calculated using statistical software MedCalc (version 22.009 for Windows). In cases when two or more consecutive samples were non-reactive by either tELISA or α-Gal-ELISA, the date of the first negative sample was considered as the time of negative seroconversion for this method. Kaplan-Meier curves and linear regression analysis were plotted and compared using Log-rank (Mantel Cox) test to obtain median time of seronegativization and ANCOVA, respectively; both available in GraphPad Prism 8 (version 8.01 for Windows; San Diego, CA, USA).

## Results

Two cohorts of Chagas disease pediatric patients were included in this study. The first cohort involved 82 *T. cruzi*-infected children (3 days to 16 years-old) from both sexes either in the

acute or the early chronic phase of Chagas disease with no evidence of cardiac abnormalities or any other disease-associated pathology at the time of diagnosis, mostly born to *T. cruzi*-infected mothers. Eighty-one out of them were treated with BZ or NF and followed up (2 to 229 months; median = 98 months) at Servicio de Parasitología-Chagas, Hospital de Niños 'Dr Ricardo Gutierrez', Buenos Aires, Argentina. A total of 509 serum samples were obtained from these patients at the time of diagnosis, treatment and follow-up (median = 6 samples per patient). This cohort was split into 3 groups based on the age of patients at the time of treatment initiation. Group 1 comprised 24 babies (3 days to 5 months-old; median = 26 days-old) that acquired *T. cruzi* infection congenitally. Group 2 comprised 31 children (7 months to 3 years-old; median = 1.0-year-old) whereas Group 3 comprised 26 patients (3 to 16 years-old; median = 8.4 years-old). The second cohort (Group 4) included 39 non-infected infants (3 days to 5 months-old; median = 31 days-old) born to *T. cruzi*-infected mothers. A total of 68 serum samples were obtained from these patients during the follow-up (median = 1 sample per patient). Samples were analyzed by tELISA and α-Gal-ELISA; and some of them were also screened for TSSA antibodies by TSSA-ELISA, and/or for the presence of the parasite or its DNA through microhematocrit and qPCR. A flow chart summarizing this information is depicted in Fig 1. The demographic, clinical and diagnostic features of each patient are summarized in S1 and S2 Tables.

## Seroprevalence for α-Gal in pediatric Chagas disease

Serological reactivity towards α-Gal in *T. cruzi*-infected patients was evaluated in samples collected at diagnosis (before treatment) by means of an α-Gal-ELISA test. Forty eight out of 82 patients yielded positive results, indicating a 58.4% seroprevalence for this glycotope. Importantly, those that presented negative α-Gal-ELISA results at diagnosis (*n* = 34), remained negative all along the follow-up (S1 Table). Stratification of patients by age, and likely by duration of infection, revealed substantial variations in α-Gal reactivity among groups. As shown, α-Gal seroprevalence was estimated at 41.7% for babies in Group 1, and 58.1% and 73.1%, respectively, for older children in Groups 2 and 3 (Table 1). One of the 82 children diagnosed with congenital *T. cruzi* infection, and showing α-Gal reactivity, did not undergo trypanocidal treatment and was not included in any group (S1 Table). The same analysis was performed in serum samples from healthy babies under 5 months born to *T. cruzi*-infected mothers (Group 4). As described for age-matched Group 1, few of these patients (9 of 39, 23.1%) displayed antibodies to α-Gal (Table 1).

The lower α-Gal seroprevalence recorded among babies under 5 months from Groups 1 and 4, in which serological reactivity is mainly shaped by maternal antibodies [3], prompted us to evaluate α-Gal reactivity in serum samples from mother/child binomials. For comparison, we also measured in these binomials the serological responses to an α-Gal unrelated *T. cruzi* antigen, TSSA. For Group 1 binomials (*T. cruzi*-infected mothers/congenitally infected newborns; *n* = 15), 15 out of 15 mothers (100%) showed reactivity in both α-Gal-ELISA and TSSA-ELISA (Table 2). However, of 15 newborns from this group, 14 (93.3%) displayed TSSA antibodies and only 7 (46.7%) yielded positive results in α-Gal-ELISA at the earliest sampling point (0.3 to 3 months after birth). A similar situation was observed for Group 4 binomials (*T. cruzi*-infected mothers/uninfected newborns; *n* = 39). In this case, out of 39 evaluated mothers, 36 (92.3%) displayed TSSA reactivity and 26 (66.7%) α-Gal reactivity. Again, the fraction of children showing antibody reactivity at the earliest sampling point (0.1 to 3.9 months after birth) was much higher for TSSA (81.1%; 31 TSSA-reactive babies born to 36 TSSA-reactive mothers) as compared to α-Gal (34.6%; 9 α-Gal-reactive babies born to 26 α-Gal-reactive mothers) (Table 2).

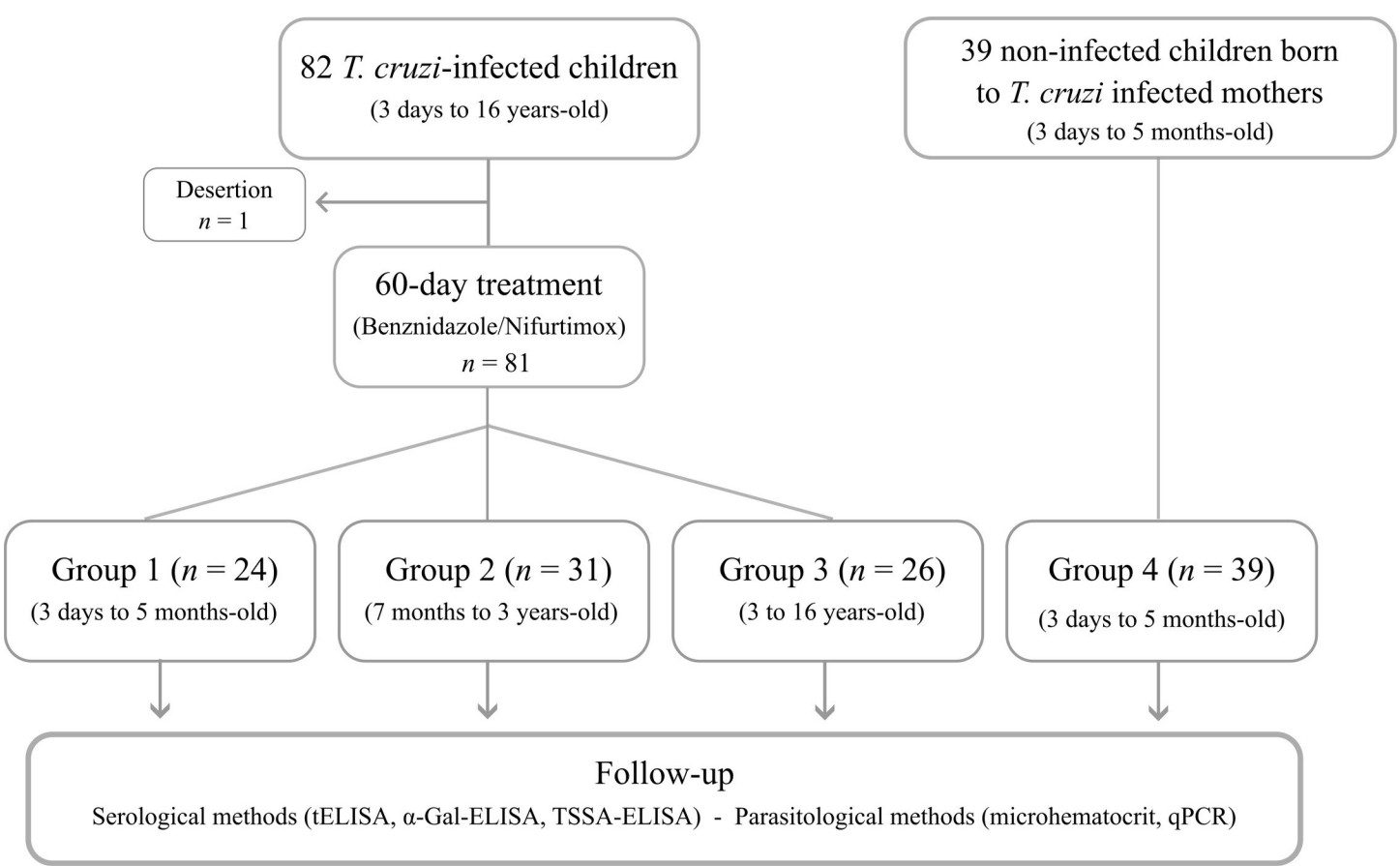

**Fig 1. Study population, inclusion criteria and group composition.**

### α-Gal antibodies and assessment of therapy efficacy

Due to methodological issues, i.e., availability of properly time-spaced samples, we only considered data corresponding to 62 out of 81 *T. cruzi*-infected and treated children for the evaluation of α-Gal antibody decay. Compilation of serological results for the whole cohort can be found at S1 Table. Fifty nine out of these 62 patients (95.2%) showed a steady decrease in tELISA reactivity after treatment, thus suggesting efficacious chemotherapy (S1 Table). Of these, only 37 patients displayed α-Gal reactivity, thus allowing the direct comparison of parasite-specific antibody responses along the follow-up. Thirty-four of them (91.9%) achieved negative seroconversion at least for one method: 18 were negativized for α-Gal-ELISA but not for tELISA, 1 for tELISA but not for α-Gal-ELISA and 15 for both techniques (Fig 2A). Interestingly,

**Table 1. α-Gal seroprevalence in pediatric Chagas disease.**

| Group | Clinical status | Age range | n | α-Gal reactive (seroprevalence) |
|---|---|---|---|---|
| 1 | *T. cruzi*- Infected | <5 months[1] | 24 | 10 (41.7%) |
| 2 | | 7 months-3 years[1] | 31 | 18 (58.1%) |
| 3 | | 3–16 years[1] | 26 | 19 (73.1%) |
| 4 | Non-infected | <5 months | 39 | 9 (23.1%) |

[1]At the time of treatment initiation.

Table 2. α-Gal antibodies in *T. cruzi* infected-mother/newborn binomials.

| Binomials | Clinical status | n | sampling point[1] | α-Gal | TSSA | tELISA |
|---|---|---|---|---|---|---|
| Group 1 | *T. cruzi*-infected mothers | 15 | | 15 (100%) | 15 (100%) | 15 (100%) |
| | *T. cruzi*-infected babies | 15 | 1.2 (0.3–3) | 7 (46.7%) | 14 (93.3%) | 15 (100%) |
| Group 4 | *T. cruzi*-infected mothers | 39 | | 26 (66.7%) | 36 (92.3%) | 39 (100%) |
| | Non-infected babies | 39 | 1.0 (0.1–3.9) | 9 (34.6%) | 31 (81.1%) | 39 (100%) |

[1]The median time after birth (and range) for the whole group is expressed in months.

negative seroconversion for the latter patients occurred either before ($n = 11$) or at the same time ($n = 4$) in α-Gal-ELISA as compared to tELISA (Fig 2A). Overall, the rate of patient negative seroconversion was significantly higher for α-Gal-ELISA as compared to tELISA (89.2% vs 43.2%, $p < 0.005$); and Kaplan-Meier curves displaying the performance of both methods are plotted in Fig 2B. As shown, the median time values of negativization were significantly shorter for α-Gal-ELISA than for tELISA (23 months [IC 95% 7 to 36 months] vs 60 months [IC 95% 15 to 83 months]; $p = 0.0016$).

Kinetics of negativization were also assessed for each group, upon stratification of patients (Fig 3). For Group 1, the median time values of negativization for α-Gal-ELISA and tELISA were 2.1 months (IC 95% 0.7 to 12.3 months) and 5.4 months (IC 95% 4.9 to 8 months), respectively (Fig 3A). For Group 2, these values were significantly different ($p = 0.0001$): 7 months (IC 95% 1.5 to 19 months) for α-Gal-ELISA and 46 months (IC 95% 19 to 83 months) for tELISA (Fig 3B). Finally, for Group 3 patients, the median time value of negativization for α-Gal-ELISA was 53 months (IC 95% 26 to 84 months), while none of them achieved negative serological conversion for tELISA (Fig 3C). Serological regression slopes were not significantly different when assessed by either tELISA or α-Gal-ELISA, neither when analyzed globally in the whole cohort, nor upon its stratification (S1 Fig).

Subsequently, we assessed the decay of antibody responses for non-infected babies from Group 4 (S2 Table). Those showing α-Gal reactivity ($n = 9$) achieved negative seroconversion for both methods, although this negativization occurred invariably before in α-Gal-ELISA as compared to tELISA (Fig 2A). The median time values of negativization for Group 4 patients were significantly shorter ($p = 0.0001$) for α-Gal-ELISA than for tELISA (3.6 months [IC 95% 0.6 to 4.8 months] vs 8 months [IC 95% 6.4 to 9.1 months]) (Fig 3D). Notably, kinetics of antibody decay for this group were very similar to those recorded for age-matched Group 1 (S2 Fig), suggesting that, if timely treated, *T. cruzi*-infected children do not elicit robust serological responses to the parasite.

## Particular cases

Three out of 62 patients considered for the evaluation of post-therapy serological responses (the 3 of them reactive for α-Gal) did not show a steady decline in tELISA reactivity after treatment and were thus analyzed separately.

**#554** (Group 2): This patient presented negative results for molecular tests and a steady decrease in anti-*T. cruzi* antibody titers early during the follow-up, suggesting an efficacious treatment (Fig 4A). Negative seroconversion for tELISA and α-Gal-ELISA was achieved at 5 months post-treatment (m.p.t.), thus well within values determined for Group 2 (Fig 3). Unexpectedly, this patient reversed to positive results for both serological methods at subsequent sampling points (11, 34 and 40 m.p.t.), suggestive of infection reactivation. It should be noted, however, that qPCR results remained negative throughout this period (Fig 4A). At 68 m.p.t., serological values resumed the decreasing trend and finally negativized.

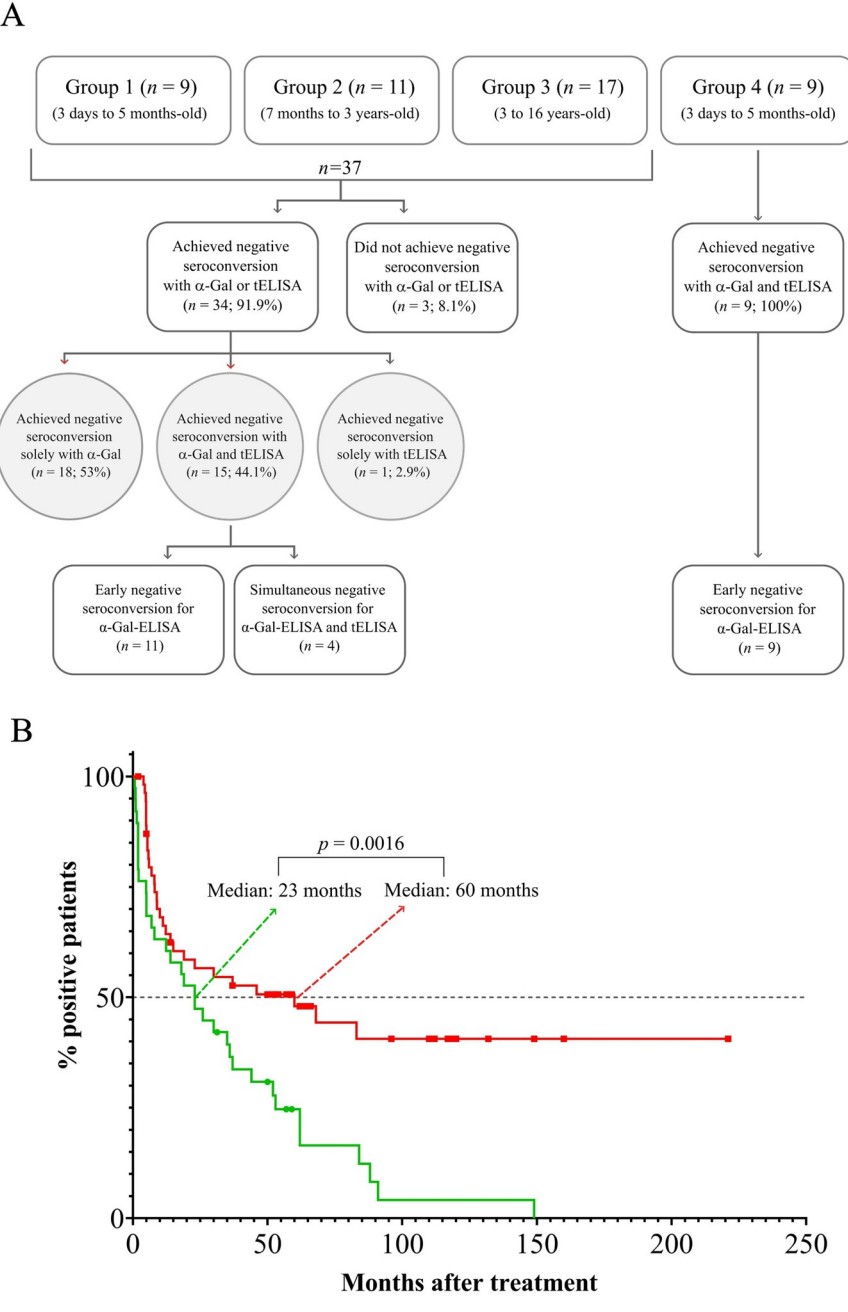

**Fig 2. Serological responses in patients with efficacious chemotherapy. A)** Diagram showing direct comparison of α-Gal-ELISA and tELISA results in the fraction (*n* = 37) of α-Gal reactive patients. **B)** Kaplan-Meier curves showing the survival time for antibodies determined by tELISA (red; *n* = 59) and α-Gal-ELISA (green; *n* = 37) in *T. cruzi*-infected children. Median negativization values are indicated for each data set. Censored cases are indicated with filled circles. Log-rank (Mantel-Cox) analyses were performed to compare median time of negative seroconversion.

**#783** (Group 3): This patient was negativized for qPCR at 1 m.p.t. A steady decay in anti-*T. cruzi* antibodies was also observed, leading to negative seroconversion for α-Gal-ELISA, but not for tELISA, at 73 m.p.t. (Fig 4B), coherent with the serological behavior of Group 3 patients. However, and as with patient #554, patient #783 displayed α-Gal reactivity at the next sampling point (82 m.p.t.). α-Gal seroconversion was confirmed at 92 and 114 m.p.t., when an

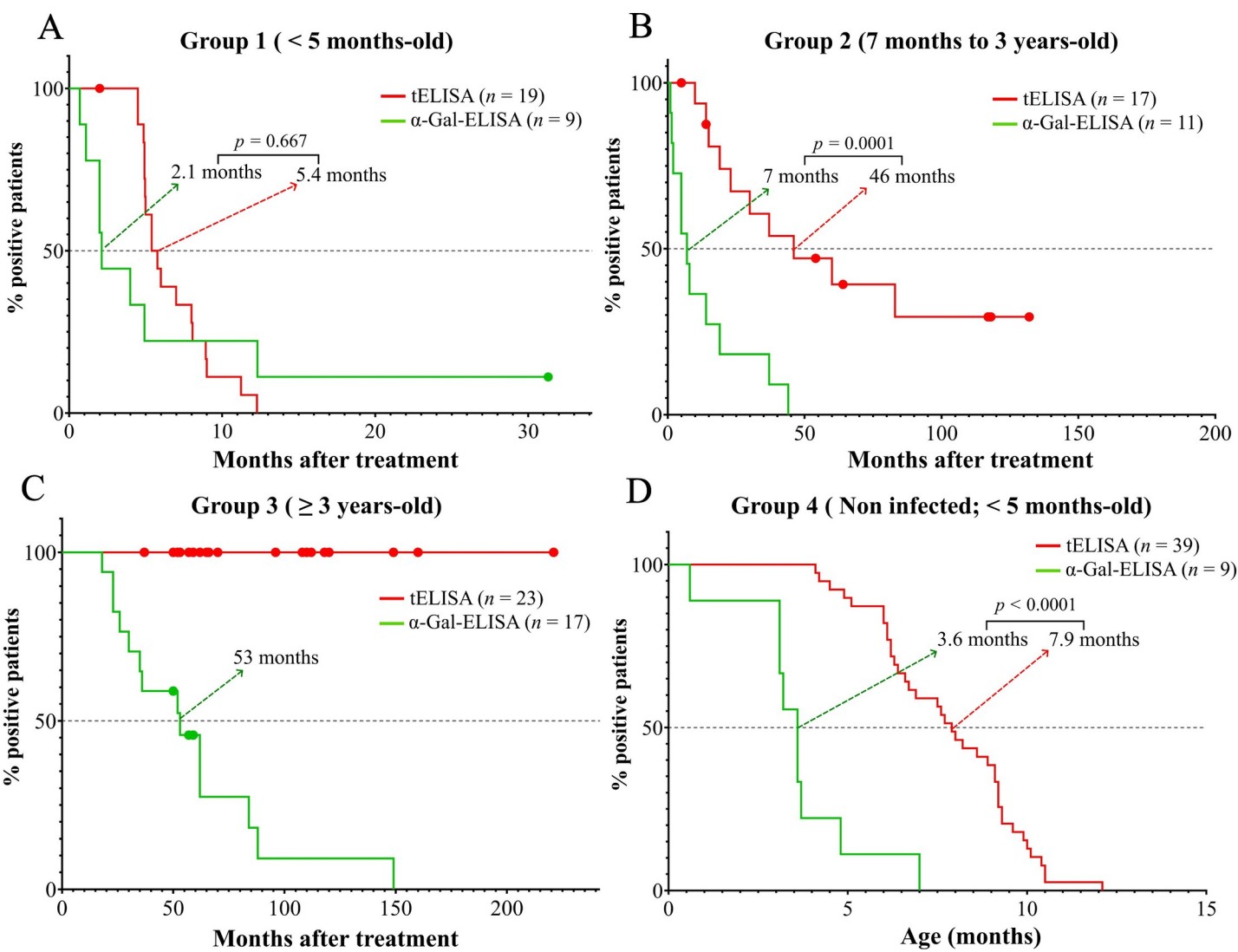

**Fig 3. Kaplan-Meier curves of patients from different groups.** Survival time of antibodies detected by tELISA (red lines) and α-Gal-ELISA (green lines) for patients from Group 1 (**A**), Group 2 (**B**), Group 3 (**C**) and Group 4 (**D**). Median values are indicated for each data set. Censored cases are indicated with filled circles. Log-rank (Mantel-Cox) analyses were performed to compare median time of negative seroconversion.

increase in tELISA reactivity was also verified (Fig 4B), thereby suggesting infection reactivation. Unfortunately, qPCR studies were not carried out at these time points. At subsequent samplings, *T. cruzi* antibody titers decayed, though not leading to α-Gal-ELISA or tELISA negativization by the end of the follow-up (152 m.p.t.).

**#1598** (Group 2): This patient was apparently infected with *T. cruzi* congenitally, as indicated by positive qPCR results at 1.5 months after birth (Fig 4C). At this time-point, serological reactivity for tELISA and α-Gal-ELISA was detected. Both kinds of antibodies rapidly decayed to baseline (4.2 months for α-Gal-ELISA and 7.2 months for tELISA), suggesting they were of maternal origin. Unexpectedly, this patient yielded positive results for both qPCR and α-Gal-ELISA at 10.5 months, suggestive of infection reactivation. These results were confirmed at the next sampling point (19.2 months), when patient #1598 also became positive for tELISA and was immediately treated (Fig 4C).

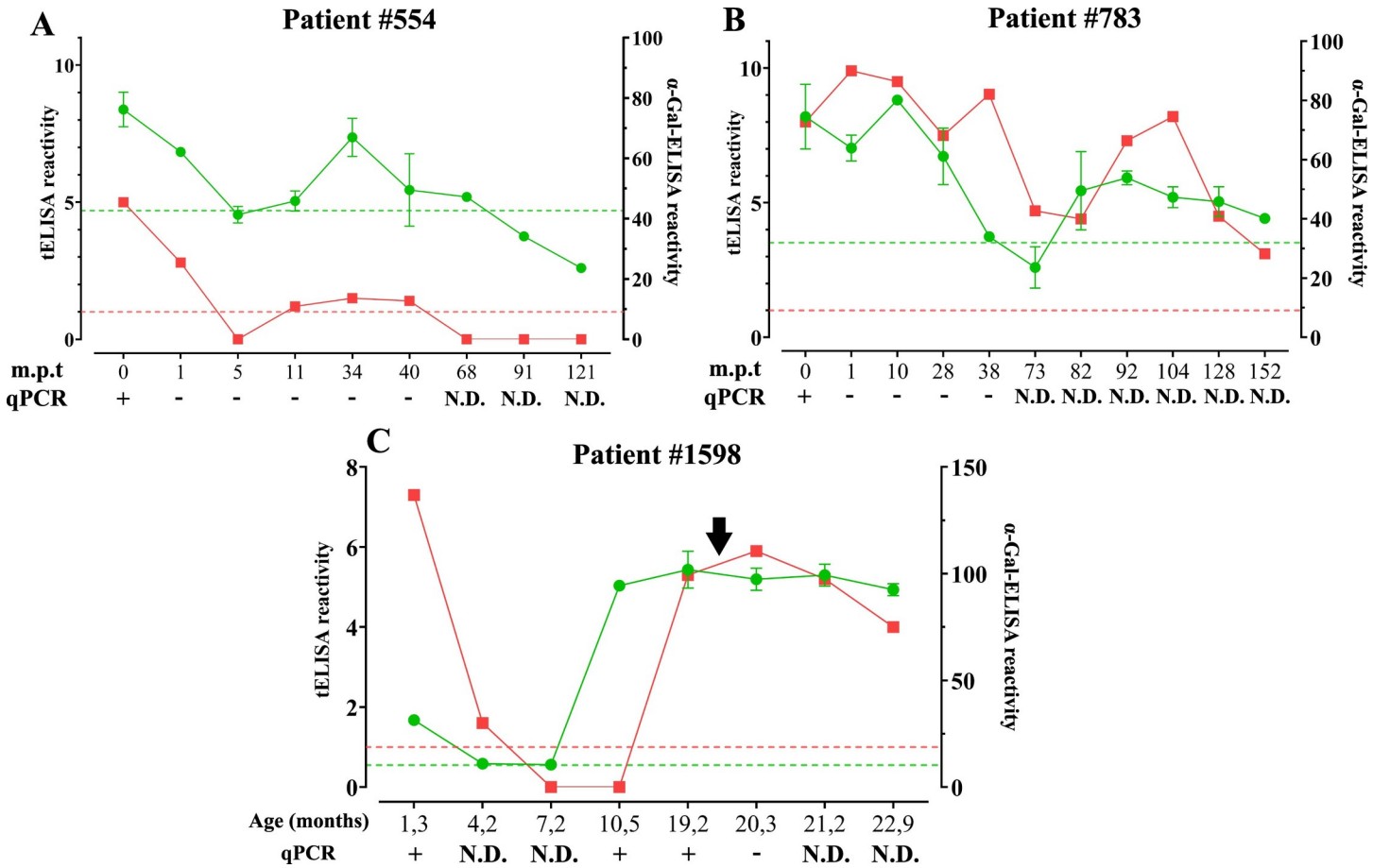

**Fig 4. Serological results for patients #783, #554 and #1598. A-C)** tELISA (red), α-Gal-ELISA (green) and qPCR results (+, positive; -, negative; N.D., non-determined) for patient #554 (**A**), #783 (**B**) and #1598 (**C**). Treatment was initiated at time 0 (expressed in months, panels **A** and **B**) or at 19 months after birth (arrow, panel **C**). The cutoff determined for tELISA and α-Gal-ELISA are indicated with color-matched dotted lines. m.p.t., months post-treatment.

## Discussion

The measurement of antibody responses to defined *T. cruzi* antigen(s) or antigenic fractions instead of crude extracts brought about significant improvements in Chagas disease serodiagnosis [15]. Among the screened candidates, the tGPI-mucins antigenic fraction demonstrated excellent performance both as a diagnostic and post-therapeutic biomarker [19,28–32]. The cost and technical difficulties associated with tGPI-mucins preparation (culture of parasite infective forms, laborious downstream purification procedures, batch-to-batch inconsistencies, etc.), however, precluded its routine implementation in clinical settings. In this context, the use of NGPs built upon tGPI-mucins glycans emerges as an appealing approach to overcome these issues. As a first step towards this goal, we have recently developed BSA-Tri, a NGP consisting of BSA surface-labeled with several units of the synthetic trisaccharide α-D-Gal*p*(1→3)-β-D-Gal*p*(1→4)-β-D-Glc*p* [39]. Further studies indicated that BSA-Tri is recognized by α-Gal antibodies obtained from *T. cruzi*-infected individuals; and that this recognition is ascribed to its terminal α-D-Gal*p*(1→3)-β-D-Gal*p* glycotope, which is shared with α-Gal [39]. Building on this knowledge, we herein evaluated an ELISA test based on this reagent (α-Gal-ELISA) for the serological follow-up of a large cohort of Chagas disease pediatric patients.

This method displayed a shorter median time value of negativization following treatment than tELISA (23 months [IC 95% 7 to 36 months] vs 60 months [IC 95% 15 to 83 months]; $p$ = 0.0016). In addition, α-Gal-ELISA led to a higher rate of patient negative seroconversion as compared to tELISA (89.2% vs 43.2%, $p < 0.005$). Moreover, direct in-patient comparative analyses indicated that negative seroconversion for α-Gal-ELISA usually precedes negativization for tELISA. The outperformance of α-Gal-ELISA as compared to conventional serology was also verified upon the stratification of patients by age (and hence most likely by duration of the infection). For Groups 1 (babies under 5 months) and 2 (7 months to 3 years-old children), for instance, the median time values of negativization were reduced for α-Gal-ELISA than tELISA. Most remarkably, of 24 patients belonging to Group 3 (3 to 16 years-old children), 14 (58.3%) achieved negative seroconversion for α-Gal-ELISA whereas none of them negativized for conventional serology. The difficulty in achieving post-treatment negativization using conventional serological methods in children aging >3 years (and also in chronically infected adults) constitutes a major issue in the clinical management of Chagas disease, and stresses the urgent need of surrogate drug efficacy indicators [7,31,44,45]. In addition to providing a novel tool to shorten follow-up periods after chemotherapy, the α-Gal-ELISA developed here may also assist *T. cruzi* infection diagnosis in certain clinical situations. Our findings with particular cases #554, #783 and #1598, though preliminary, suggested that this method is a better indicator of infection reactivation and/or reemergence than conventional serology.

The main limitation of the α-Gal-ELISA is related to its suboptimal sensitivity. The seroprevalence in our pediatric cohort was estimated at 58.4%, indicating that a substantial proportion of patients would not benefit with its use. The fact that this seroprevalence is lower than those reported for parasite-derived tGPI-mucins in previous studies carried out on Chagas disease pediatric populations may be attributed to variations in the clinical, immunological, and/or immunogenetic features of patients [29–32]. Alternatively, and most likely, this discrepancy may reflect variations in the antigenic constitution of the used reagents, NGP-Tri vs tGPI-mucins. As mentioned, tGPI-mucins is a poorly defined antigenic fraction, made up of highly hydrophilic, surface glycoconjugates obtained from *T. cruzi* trypomastigote forms. Even though its reactivity is mainly driven by α-Gal structures, the existence of additional peptide and/or glycan epitopes on tGPI-mucins has been described [19,20,46–48]. Moreover, depending on the tGPI-mucins batch preparation, and particularly if the final octyl-Sepharose purification step is omitted, its contamination with other highly antigenic molecules from the parasite cannot be ruled out.

Interestingly, the α-Gal seroprevalence determined in this study was not homogeneous across groups but rather biased by a high proportion of non-responders in infants under 5 months of age. In the search for an explanation to this finding, and considering that serological reactivity in these patients is mainly shaped by maternal antibodies [3], we analyzed α-Gal responses in *T. cruzi*-infected mother/infant binomials. For comparison, we assessed antibodies to a well-characterized and immunodominant *T. cruzi* peptide antigen, TSSA [42,43]. As shown, most of the babies (~81–93%) born to TSSA-reactive mothers displayed TSSA antibodies whereas only a minor fraction (~34–47%) of those born to α-Gal-reactive mothers yielded positive results in α-Gal-ELISA. Therefore, we hypothesize that the lower-than-expected α-Gal reactivity in infants under 5 months of age might be due to a suboptimal mother-to-infant transmission of α-Gal antibodies and/or to the extremely fast clearance of α-Gal maternal antibodies in newborns. Importantly, α-Gal seroprevalence in babies born to *T. cruzi*-infected mothers seems irrespective of their infection status, as it was observed both in Group 1 (congenitally infected) and Group 4 (non-infected) patients. When only children >7 months (Groups 2 and 3) and/or patients chronically infected with *T. cruzi* (mothers) were

considered, α-Gal seroprevalence increased up to ~70–80%, well in the range of other *T. cruzi* antigens proposed as post-therapeutic biomarkers for Chagas disease [18,44].

Overall, our α-Gal-ELISA test showed a very good performance on the cure assessment of Chagas disease in pediatric populations and may also be useful for other diagnostic needs. These results warrant further studies, particularly aimed to increase the sensitivity of the method. In this line, the implementation of highly sensitive detection systems, i.e. fluorescence or chemiluminescence-based techniques and/or the incorporation of additional, tGPI-mucins α-galactosyl-based glycotopes inferred by reverse immunoglycomics [38] to our diagnostic reagent will be explored. Most importantly, implementation of the tools developed here (and their optimized versions) is expected to have a positive impact on the clinical management of Chagas disease as well as on the identification/validation of novel drug candidates for the treatment of *T. cruzi* infections.

## Supporting information

**S1 Table. Clinical and diagnostic features of patients from Groups 1 to 3.** (XLSX)

**S2 Table. Clinical and diagnostic features of patients from Group 4.** (XLSX)

**S1 Fig. Serological regression analyses.** Serological profiles for tELISA and α-Gal-ELISA for patients from Group 1 (A), Group 2 (B), Group 3 (C) and the whole cohort of T. cruzi-infected children (D). Reactivity values are expressed as % of the first (pre-treatment, P) sample and regression curves are indicated in red and green lines, respectively. Mean reactivity and SD values for each time point are shown in red (tELISA) and green (α-Gal-ELISA) dots. Slope (95% CI) and $R^2$ values are indicated for each data set. ANCOVA analyses were performed to compare slopes. (TIF)

**S2 Fig. Babies born to T. cruzi-infected mothers and congenitally infected (Group 1) or not (Group 4) with the parasite show similar kinetics of decay of anti-parasite antibodies.** Kaplan-Meier curves showing the survival time for antibodies detected by tELISA (red) and α-Gal-ELISA (green) for patients from Group 1 (dashed lines) and Group 4 (solid lines). Median negativization values are indicated for each data set. Censored cases are indicated with dots. Log-rank (Mantel-Cox) analyses were performed to compare median time of negative seroconversion. (TIF)

## Acknowledgments

This paper is dedicated to the memory of Dr A.C.C. Frasch, co-founder and first director of the IIBio.

## Author Contributions

**Conceptualization:** Virginia Balouz, Jaime Altcheh, Carlos A. Buscaglia.

**Data curation:** Cintia V. Cruz, Jaime Altcheh.

**Formal analysis:** Manuel Abal, Virginia Balouz, M. Eugenia Giorgi, Carla Marino, Cintia V. Cruz, Jaime Altcheh.

**Funding acquisition:** Virginia Balouz, Carla Marino, Jaime Altcheh, Carlos A. Buscaglia.

**Investigation:** Manuel Abal, Virginia Balouz, Rosana Lopez, M. Eugenia Giorgi, Carla Marino, Cintia V. Cruz.

**Methodology:** Manuel Abal, Virginia Balouz, Rosana Lopez, M. Eugenia Giorgi, Carla Marino, Cintia V. Cruz.

**Project administration:** Jaime Altcheh, Carlos A. Buscaglia.

**Resources:** Rosana Lopez, M. Eugenia Giorgi, Carla Marino.

**Supervision:** Virginia Balouz, Jaime Altcheh, Carlos A. Buscaglia.

**Writing – original draft:** Carlos A. Buscaglia.

**Writing – review & editing:** Manuel Abal, Virginia Balouz, M. Eugenia Giorgi, Carla Marino, Cintia V. Cruz, Jaime Altcheh, Carlos A. Buscaglia.

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
