## [Decision Letter · Decision Letter 0]

2 Jan 2024

Dear Dr Buscaglia,

Thank you very much for submitting your manuscript "An α-Gal antigenic surrogate as a biomarker of treatment evaluation in Trypanosoma cruzi-infected children. A retrospective cohort study" for consideration at PLOS Neglected Tropical Diseases. As with all papers reviewed by the journal, your manuscript was reviewed by members of the editorial board and by several independent reviewers. The reviewers appreciated the attention to an important topic. Based on the reviews, we are likely to accept this manuscript for publication, providing that you modify the manuscript according to the review recommendations. 

Sincerely,

Charles L. Jaffe, Ph.D.

Section Editor

Reviewer's Responses to Questions

**Key Review Criteria Required for Acceptance?**

**Methods**

-Are the objectives of the study clearly articulated with a clear testable hypothesis stated?

-Is the study design appropriate to address the stated objectives?

-Is the population clearly described and appropriate for the hypothesis being tested?

-Is the sample size sufficient to ensure adequate power to address the hypothesis being tested?

-Were correct statistical analysis used to support conclusions?

-Are there concerns about ethical or regulatory requirements being met?

Reviewer #1: The objectives of the study are clearly formulated in both the Abstract and Introduction; the study design is concordant with these objectives. The study aimed at evaluating serological responses to a neoglycoprotein consisting of a BSA scaffold decorated with several units of a synthetic α-Gal antigenic surrogate (α-D-Galp(1→3)-β-D-Galp(1→4)-D-Glcp) in a population of pediatric Chagas disease treated with benznidazole (BZ) or nifurtimox (NF) and was carried out in a cohort of 82 children (3 days to 16 years-old at the time of treatment initiation). Statistical analysis of the results was carried out as described in detail in Methods.

Reviewer #2: The answer to all questions is yes.

Reviewer #3: (No Response)

**Results**

-Does the analysis presented match the analysis plan?

-Are the results clearly and completely presented?

-Are the figures (Tables, Images) of sufficient quality for clarity?

Reviewer #1: The analysis of the results is concordant with the analysis plan. Key results are that α-Gal-ELISA yielded shorter median time values of negativization (23 months [IC 95% 7 to 36 months] vs 60 months [IC 95% 15 to 83 months]; p = 0.0016) and higher rate of patient negative seroconversion (89.2% vs 43.2%, p < 0.005) as compared to conventional serological methods, but has a suboptimal seroprevalence (58.4%) among tested children. To address this serious limitation the authors suggest the implementation of highly sensitive detection systems, i.e. fluorescence or chemiluminescence-based techniques and/or the incorporation of additional, F2/3 α-galactosyl-based glycotopes inferred by reverse immunoglycomics to their diagnostic reagent.

Reviewer #2: The answer to the first two questions is yes.

The figures appear to be of low quality (not as sharp as they could be). This can probably be fixed by changing the format.

Reviewer #3: Yes, no objections

**Conclusions**

-Are the conclusions supported by the data presented?

-Are the limitations of analysis clearly described?

-Do the authors discuss how these data can be helpful to advance our understanding of the topic under study?

-Is public health relevance addressed?

Reviewer #1: The conclusions are consistent with the data presented and the limitations of the new diagnostic reagent are clearly acknowledged. The results and conclusions are definitely relevant for the diagnosis of Chagas disease and the evaluation of the efficacy of treatments.

Reviewer #2: The answer to all four questions is yes.

Reviewer #3: yes, the MS meets all these criteria

**Editorial and Data Presentation Modifications?**

Reviewer #1: The authors should include an additional bibliographic reference to the development of neoglycoproteins containing synthetic α-Gal antigenic moieties of Chagas disease diagnosis: Ortega-Rodriguez et al. 2019. Meth Mol Biol. chapter 22.

Reviewer #2: Minor Revision

Reviewer #3: Text revision: I am providing a brief list of comments below

**Summary and General Comments**

Reviewer #1: The work is highly relevant for the development of new, improved diagnostic methods for the diagnosis of Chagas disease and evaluation of treatments' efficacy. The article deserves publication in PLOS Neglected Tropical Diseases.

Reviewer #2: Buscaglia and coworkers describe the use of the synthetic neoglycoprotein BSA-Tri (consisting of BSA with multiple copies of the trisaccharide α-D-Galp(1→3)-β-D-Galp(1→4)-D-Glc conjugated to it), as a biomarker for chemotherapy efficacy when treating Chagas disease (CD) in babies and children. These patients were treated with benznidazole or nifurtimox, the only two drugs available for CD. Parasitological methods and PCR are not suitable to determine clearance of the parasite, which also hampers the development of new anti-CD drugs. The only acceptable method to determine cure from CD is negative antibody seroconversion, which requires a suitable antigen. T. cruzi trypomastigote lysates are not suitable for the serological determination of treatment efficacies due to the slow antibody seroconversion, which takes years. An alternative antigen is the T. cruzi derived F2/3 glycoprotein fraction, which is rich in terminal, highly immunogenic and antigenic α -Gal residues. The F2/3 can be used for the serological follow-up after treatment by monitoring a patients anti-alpha-Gal Ab levels, but the preparation of F2/3 from trypomastigotes is very challenging and not feasible on a large scale. Since currently no molecular carbohydrate biomarker is available, the authors used the above mentioned BSA-Tri for determining the treatment efficacy. Using α-D-Galp(1→3)-β-D-Galp(1→4)-D-Glc as a surrogate for F2/3 is a clever antigen choice because it contains the immunodominant glycotope α-D-Galp(1→3)-β-D-Galp, recognized by anti-alpha-Gal Abs of CD patients, and it is synthetically more accessible than the T. cruzi cell surface glycan α-D-Galp(1→3)-β-D-Galp(1→4)-D-GlcNAc. Before and after treatment, the authors measured the patients' anti-alpha-Gal levels by ELISA and found that seroconversion was significantly faster using BSA-Tri than conventional serology. Also, there was a trend that indicated that younger children seroconverted faster than older children. Despite the fact that the seroprevalence was rather low (only 58.4% of the children showed an Ab response to BSA-Tri) most likely due to sensitivity issues of the ELISA, this biomarker could proof useful in determining treatment efficacy in pediatric CD. Testing the performance of BSA-Tri is novel The manuscript is well written, the findings are significant and are of great interest to the readership of PLOS ntd. The In the opinion of this reviewer, the manuscript is suitable for publication in PLOS ntd after minor revisions.

Criticism:

1. The anomeric stereochemistry of the Glc units in BSA-Tri should be indicated. Is it alpha or beta?

2. Please comment on how it was determined that the second cohort, the babies of CD mothers, were indeed not infected with T. cruzi. If some of them were in reality T. cruzi positive, how will this affect data interpretation?

3. On page 6, please reference (a) Almeida et al., J. Clin. Lab Anal. 1993, DOI: 10.1002/jcla.1860070603; which demonstrates F2/3 as therapeutic marker in CD.

4. At the bottom of page 6, three references should be added, which suggest using T. cruzi-derived NGPs as prognostic biomarkers for chemotherapy follow-up:

(a) Ortega-Rodriguez, U. et al. Methods Mol. Biol. 2019; doi: 10.1007/978-1-4939-9148-8_22.

(b) Montoya et al, Molecules 2022, https://doi.org/10.3390/molecules27175714. (This publication is referenced later in the manuscript, but should be referenced here already.)

(c) Alonso-Vega, C. et al., BMJ Open, 2021, doi:10.1136/ bmjopen-2021-052897

Reviewer #3: This is a relevant serological study aiming to define whether chagasic patients were cured or not by currently available drugs (benznidazol and nifurtimox). It is well established that conventional serology cannot distinguish patients that are chronically infected from those that were cured following chemotherapy. In past studies, two groups proclaimed that cured chagasic patients could be discriminated by measuring clearance rates of protective IgG antibodies directed against alfa galactose glycotope, either expressed in F2;3 surface antigens (authors previous work) or T. cruzi mucins (Almeida's group). Avoiding the complexity of antigen purification and batch variability of F2-3, here Buscaglia et al. screened sera from different age groups of pediatric cohorts (N=82 3-16 years old) using BSA coupled to a previously defined alfa-gal glycotope termed NGP-tri. The rate of clearance of IgG antibodies to NGP-tri was compared with antibodies directed to another well-characterized T. cruzi antigen (TSSA). Different curves were generated as a function of the children age, separated in 4 groups. Although the sensitivity of antibody detection in screens performed with NGP-tri is till not ideal for pediatric cases (the authors conceded that the technique can be improved), negative seroconversion was achieved in 23 months and 89.2% of the cases as compared to conventional serology (60 months, 43.2% of the patients). Moreover, the trend of negativization for NGP-tri was faster as compared to rates of TSSA specific antibodies. 

Suggestions

Discussion

You could devote a paragraph to express your thoughts about the results obtained in 

"exceptional" cases. It is not clear why patient 1598 is PCR positive for T cruzi at 1.3 months and only treated at 19 months. Please address the obvious possibility that the early drop of alfa-Gal ELISA responses of this patient reflect competition of IgA serum antibodies (NGP-tri or TSSA) acquired during breast feeding. Microbiota is changing. Gradual elevation of serum IgA levels could even interfere with the early drop observed in 554 and 783, both treated with antiparasite drugs at time zero. From a perspective of basic science, this information (serum IgA level NGP-tri and TSSA antigens) would add interest to the work (readers of PLOS Neglected Trop Diseases have broader interests than clinical journals). 

References-There is an impressive number of self citations- please make sure that the proposition to use alga-gal glycopeptides from T. cruzi mucins as biomarkers of cure is properly cited both in the introduction and discussion. Making justice to the history of progress in each field is always important.

Minor problems

Introduction 

line 84- most important parasitic disease in Latin America- overstatement, please change the sentence- not justified anymore. 

line 95-add reference after lower and variable. 

line 98- their uncertain teratogenic risk. Suffice to write due to teratogenic risk

line 102- the only "accepted" criteria of cure is... - you must add a reference-either WHO or other health authority. Otherwise, change the sentence.

line 109. measure of parasite elimination - criteria would be better

line 126- inactivation of the alfa 1-3 galatosyltransferase gene (please describe the genetic change)

line 135- post therapeutic marker is not what you mean. Perhaps change for bker of cure...

Methods.

line 168-forgot to inform sex distribution

line 70. We are not informed how many were treated with BZN versus Nifurtimox

line 211- be more precise- aforementioned is too economical

explain the criteria used to define group 4 (39 children) as non infected.

PLOS authors have the option to publish the peer review history of their article (what does this mean?). If published, this will include your full peer review and any attached files.

Reviewer #1: Yes: Julio A. Urbina

Reviewer #2: No

Reviewer #3: Yes: Julio Scharfstein

Figure Files:

Data Requirements:

Please note that, as a condition of publication, PLOS' data policy requires that you make available all data used to draw the conclusions outlined in your manuscript. Data must be deposited in an appropriate repository, included within the body of the manuscript, or uploaded as supporting information. This includes all numerical values that were used to generate graphs, histograms etc.. For an example see here: http://www.plosbiology.org/article/info:doi%2F10.1371%2Fjournal.pbio.1001908#s5.

Reproducibility:

References

---

## [Editor Report · Decision Letter 1]

9 Jan 2024

Dear Dr Buscaglia,

We are pleased to inform you that your manuscript 'An α-Gal antigenic surrogate as a biomarker of treatment evaluation in Trypanosoma cruzi-infected children. A retrospective cohort study' has been provisionally accepted for publication in PLOS Neglected Tropical Diseases.

Before your manuscript can be formally accepted you will need to revise three minor points raised by one of the editors (see "Academic Editor's Comments" below) and complete some formatting changes, which you will receive in a follow up email. A member of our team will be in touch with a set of requests.

Best regards,

Igor C. Almeida

Academic Editor

Charles Jaffe

Section Editor

ACADEMIC EDITOR'S COMMENTS TO THE AUTHORS:

This study by Abal et al. proposes an alpha-Gal neoglycoprotein containing multiple α-D-Galp(1→3)-β-D-Galp(1→4)-D-Glcp trisaccharide linked to BSA (NGP-Tri), in an ELISA, as a surrogate biomarker of the T. cruzi trypomastigote-derived glycosylphosphatidylinositol-anchored mucins (tGPI-MUC) for the early assessment of nifurtimox (NF) or benznidazole (BZ) chemotherapy outcomes in infected children with acute or early chronic Chagas disease (CD). Overall, this is a well-done study and confirms previous findings (see below) using tGPI-MUC to detect anti-alpha-Gal Abs in children with acute or early chronic CD, now using the synthetic α-Gal-containing NGP-Tri.

tGPI-MUC was the first biomarker (BMK) for early evaluation of cure (or failure) in a clinical study of children treated with benznidazole published in 1996 (PMID 8937280). As a matter of fact, tGPI-MUC was first shown to be a reliable marker of cure/failure following treatment in a study published in 1993 (PMID 8277354). Since then, two clinical studies in adults treated with BZN (PMID 29352704 and 33836161) using tGPI-MUC corroborated early findings by the same group and cols. Remarkably, in this current study, despite the low sensitivity (∼58%) of the α-Gal-ELISA test before treatment, which could be due to the reducing end-modified trisaccharide Galpα1,3Galβ1,4Glc[β] employed (instead of the canonical tGPI-MUC trisaccharide Galpα1,3-Galβ1,4-GlcNAcα) and/or low maternal IgG titers in the infected children, the authors find the same percentage (58%) of anti-alpha-Gal Ab seroconversion >3 years of follow-up, as previously described for the tGPI-MUC (AT Antigen) in a chemiluminescent ELISA (PMID 8937280).

This is a very nice contribution to the field of CD chemotherapy follow-up that corroborates the usefulness of anti-α-Gal Abs as reliable BMKs for premature evaluation of treatment outcomes.

Minor points (REVISION REQUIRED FOR FORMAL ACCEPTANCE):

1. Abstract: Please briefly describe the cohorts and chemotherapy regimens (NF or BZ) used.

2. Abstract, line 124, and throughout the manuscript: To avoid additional confusion in the nomenclature, please change the name of "F2/3 preparation or fraction” to TcMUC II (PMID 14749325) or tGPI-mucins (PMID 10455266, 11207300, 22764297, 30868536, 30911415), or tGPI-MUC (PMID 36080480). F2/3 preparation has the same amino acid, glycan, and GPI composition as TcMUC II or tGPI-mucins.

3. Lines 452-458: Please revise or moderate the three assertions, as they could be misinterpreted. As previously mentioned, the purified F2/3 fraction (following octyl-Sepharose) has a well-established composition, which is in close agreement with the amino acid profile of TcMUC II (refer to Table 1 in Almeida et al., Biochemical J, 1994; and Table 1 in Buscaglia et al., JBC 2004). If the F2/3 fraction were significantly contaminated with TS and MASP proteins, its amino acid composition would likely differ from that of TcMUC II. However, It is worth noting that before octyl-Sepharose purification, the F2/3 fraction may contain impurities. These contaminants are typically removed in the column’s flow-through (Vo).

---

## [Editor Report · Acceptance letter]

15 Jan 2024

Dear Dr Buscaglia,

We are delighted to inform you that your manuscript, "An α-Gal antigenic surrogate as a biomarker of treatment evaluation in Trypanosoma cruzi-infected children. A retrospective cohort study," has been formally accepted for publication in PLOS Neglected Tropical Diseases.

Best regards,

Shaden Kamhawi

co-Editor-in-Chief

Paul Brindley

co-Editor-in-Chief
